# Conversion of Green Methanol to Methyl Formate

**Doreen Kaiser, Luise Beckmann, Jan Walter**  **and Martin Bertau \***

Institute of Chemical Technology, TU Bergakademie Freiberg, Leipziger Straße 29, 09599 Freiberg, Germany; doreen.kaiser@chemie.tu-freiberg.de (D.K.); luise.beckmann@chemie.tu-freiberg.de (L.B.); jan.walter@chemie.tu-freiberg.de (J.W.)
\* Correspondence: martin.bertau@chemie.tu-freiberg.de; Tel.: +49-3731-392384

**Abstract:** Methyl formate is a key component for both defossilized industry and mobility. The current industrial production via carbonylation of methanol has various disadvantages such as high requirements on reactant purity and low methanol conversion rates. In addition, there is a great interest in replacing the conventional homogeneous catalyst with a heterogeneous one, among other things to improve the downstream processing. This is why new approaches for methyl formate are sought. This review summarizes promising approaches for methyl formate production using methanol as a reactant.

**Keywords:** carbonylation; green chemistry; methanol; methyl formate; synfuel

## 1. Introduction

Methyl formate (MF) is one of the most important building blocks in C1 chemistry and it is of great interest as an emission reducing fuel additive. MF can be synthesized from CO and methanol and can be understood as chemical storage for CO. If methanol is produced from $CO_2$ and hydrogen, which is generated through water electrolysis by renewable energy, an access to green MF is created. This way a (partial) substitution for classical i.e., fossil feedstock borne petrochemicals can be realized while reducing $CO_2$ emissions. Since 1925, MF has been produced by carbonylation of methanol with sodium methanolate as a catalyst on an industrial scale. With a global production capacity of >6 million tons in 2016 [1]. This is an industrial chemical of major interest. However, development has gone on since, and there are alternatives to the sodium methanolate process that may be more efficient or economically more attractive, or they are more sustainable.

## 2. Why Convert Methanol into Methyl Formate?

MF is one of the most important industrial products and has been widely used for the production of more than 50 chemicals, including formic acid, *N*,*N*-dimethylformamide (DMF), formamide and dimethyl carbonate (DMC) (Figure 1). Particularly, the synthesis of formic acid is of great importance. In 2016, worldwide production of the acid was estimated to be 621,000 t/a, whereby about 80% was produced by hydrolysis of MF (Kemira-Leonard process). Formic acid is largely used in pharmaceutical, food and textile industry [2].

Formamide and DMF are synthesized by the reaction of MF with ammonia or *N*,*N*-dimethylamine, respectively. Formamides perform as extraction solvents and are applied as solvents for inorganic salts, especially in polymer chemistry. Being dipolar aprotic solvents, they are ideal for nucleophilic substitutions. Therefore, DMF and formamide are mainly directly used by the manufactures [3]. MF can also react with olefins or halogenated compounds to form esters via hydroesterification or alkoxyacarbonylation reactions [4]. Trichloromethyl carbonochloridate (diphosgene) is produced by radical chlorination of MF under UV light. It was originally developed as a pulmonary agent for chemical warfare. Today it is used as an alternative to phosgene, which is easier and safer to handle. Thus, diphosgene has replaced phosgene in carbonates, polyurethane and isocyanate production

processes, which act as starting materials for high value plastics, resins, plant protection products and insecticides. Furthermore, it is used in medical field for synthesize sulfonic acid methyl pyrimidine, sulfamonomethoxine, antitussive dextromethorphan and other drugs [1].

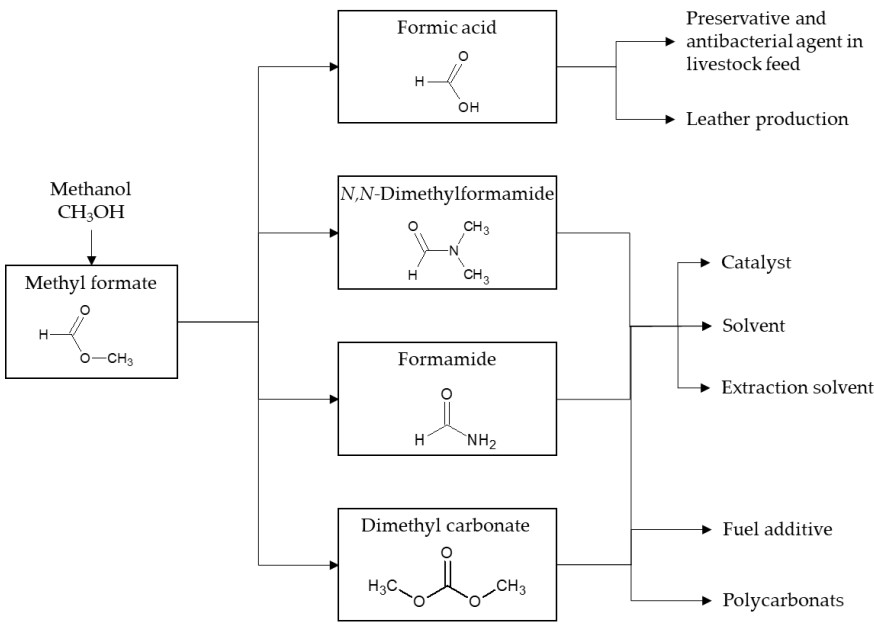

**Figure 1.** Selection of important products derivable from MF as a $C_1$ building block and their application [5].

Pure MF is known as a refrigerant (R611), used as a blowing agent for polymers, as a binder in foundry process, as a solvent for nitrocellulose as well as cellulose acetate and can be used as a smoke fumigant and bactericide for treating tobacco, dried fruit and grains. Furthermore, it is used in the pharmaceutical industry for producing sulfonic acid methyl pyrimidine, sulfamonomethoxine, antitussive dextromethorphan and other drugs [1].

A further application is fuel additive. Its research octane number (RON) and the motor octane number (MON) are similarly high (RON = 115, MON = 114.8, Table 1) [6]. The addition of MF to fuel increases knock resistance and indirectly the efficiency of combustion process. When blended with diesel, MF lowers the cloud point and avoids sedimentation of diverse fuel components at low temperatures. This improves the filtration and cold start behavior, as well as delays aging [6,7]. Moreover, with MF serves to reduce emissions of nitrogen oxides ($NO_x$) and soot. Most importantly, MF shows less toxicity compared to other hydrocarbons in gasoline or considered fuel additives [6]. According to the Global Harmonized System (GHS), methanol and higher hydrocarbons are classified as acute or organ-specific toxins. High protective measures are prescribed by law for handling, which reduces the acceptance as a fuel additive. In contrary, MF is only classified as harmful (GHS 07). Furthermore, MF does not participate in anti-atmospheric photochemical reactions and thus does not contribute to the formation of ground-level ozone and smog [8].

**Table 1.** Properties of different fuels [7].

|  | Gasoline | Methanol | Methyl Formate |
|---|---|---|---|
| RON | 97.7 | 108.7–115 | 115 |
| MON | 89 | 88.6 | 114.8 |
| $\rho$ [kg/m$^3$] | 720–780 | 800 | 957 |
| $T_{Boiling}$ [°C] | 25–210 | 64.6 | 31.5 |
| Flash Point [°C] | −40 | 12 | −19 |
| Lower heating value (LHV) [MJ/kg] | 44 | 19.7 | 15.8 |

### 3. Synthesis Routes

There are a variety of synthesis routes to produce methyl formate. On an industrial scale, the carbonylation of methanol plays the most important role. On a smaller scale, dehydrogenation and oxidation of methanol are carried out. Further possibilities are the esterification of methanol and formic acid, dimerization of formaldehyde, hydrogenation-condensation with methanol and synthesis from synthesis gas as well as photocatalysis approaches. In the following, the individual possibilities are discussed, with the focus on methanol conversion methods.

*3.1. Methanol Carbonylation*

3.1.1. Established Process

Liquid-phase carbonylation of methanol is the common approach to produce MF on an industrial scale. It was firstly industrialized by BASF in 1925. For this approach sodium methoxide (2.5 wt.%) is used as a homogeneous catalyst to synthesize MF at 80 °C and 4.5 MPa (Equation (1)) [1,4,5].

$$CH_3OH + CO \longrightarrow HCOOCH_3 \qquad \Delta H = -29.3 \text{ kJ mol}^{-1} \qquad (1)$$

Methanol carbonylation basically may yield two different products, acetic acid or MF depending on the catalyst used and the locus of CO insertion. For MF production insertion into the carbinolic O-H bond is necessary [5]. This approach is characterized by a high selectivity for MF. However, there exists a number of drawbacks: (1) sodium methoxide is sensitive towards moisture, (2) corrosion because of high alkalinity of sodium methoxide, (3) high investment costs for high pressure equipment (4), low methanol conversion (30%), (5) no separation of the catalyst and (6) low solubility of sodium methoxide. Especially, the most common catalyst itself, sodium methoxide, presents a number of problems. The high sensitivity towards water requires high purities of the raw material. The water content in methanol and the content of water, $CO_2$, $O_2$ and sulfide in CO gas must be less than 1 ppm. Furthermore, above a certain methanol concentration sodium methoxide precipitate and block pipes and valves [1,9]. In contrast to the low methanol conversion CO conversion is 95%. The yield in terms of CO and methanol is 99% and per hour and liter reaction space 800 g of MF are formed [5]. A nearly quantitative methanol conversion can be achieved by recycling the unreached methanol [10].

3.1.2. Mechanism

The methanol carbonylation occurs through a two-step mechanism. First, alcoholate reacts via nucleophilic attack with CO to form methoxycarbonyl anion ($CH_3OCO^-$) (Equation (2)) which react in the second step with methanol to methyl formate (Equation (3)). During second step the active catalyst methanolate is recovered [4,10,11].

$$O{=}C{:} \ + \ H_3C{-}O^- \ \rightleftharpoons \ H_3C{-}O{-}\overset{\overset{\displaystyle O}{\|}}{C}{}^- \qquad (2)$$

$$H_3C{-}O{-}\overset{\overset{\displaystyle O}{\|}}{C}{}^- \ + \ H_3C{-}OH \ \rightleftharpoons \ H_3C{-}O{-}\overset{\overset{\displaystyle O}{\|}}{C}H \ + \ H_3C{-}O^- \qquad (3)$$

3.1.3. By-Products

A possible side reaction is the formation of DME (Equation (4)) and TME (Equations (5) and (6)) by reaction of methoxide with MF. Furthermore, reactions of sodium methoxide with traces of water and $CO_2$ (Equations (7)–(9)) are possible and form insoluble sodium methoxycarboante [11,12].

$$\text{H}_3\text{C}-\text{O}^- \; + \; \underset{\text{H}_3\text{C}-\text{O}-\overset{\overset{\displaystyle \text{O}}{\|}}{\text{CH}}}{} \; \rightleftharpoons \; \text{H}_3\text{C}-\text{O}-\text{CH}_3 \; + \; \underset{\text{O}^-\!-\overset{\overset{\displaystyle \text{O}}{\|}}{\text{CH}}}{} \tag{4}$$

$$\text{H}_3\text{C}-\text{O}^- \; + \; \underset{\text{H}_3\text{C}-\text{O}-\overset{\overset{\displaystyle \text{O}}{\|}}{\text{CH}}}{} \; \rightleftharpoons \; \text{H}_3\text{C}-\text{O}-\overset{\overset{\displaystyle \text{O}^-}{|}}{\underset{\underset{\displaystyle \text{O}-\text{CH}_3}{|}}{\text{CH}}} \tag{5}$$

$$\text{H}_3\text{C}-\text{O}-\overset{\overset{\displaystyle \text{O}^-}{|}}{\underset{\underset{\displaystyle \text{O}-\text{CH}_3}{|}}{\text{CH}}} \; + \; \underset{\text{H}_3\text{C}-\text{O}-\overset{\overset{\displaystyle \text{O}}{\|}}{\text{CH}}}{} \; \rightleftharpoons \; \text{H}_3\text{C}-\text{O}-\overset{\overset{\displaystyle \text{O}-\text{CH}_3}{|}}{\underset{\underset{\displaystyle \text{O}-\text{CH}_3}{|}}{\text{CH}}} \; + \; \underset{\text{O}^-\!-\overset{\overset{\displaystyle \text{O}}{\|}}{\text{CH}}}{} \tag{6}$$

$$\text{CH}_3\text{ONa} + \text{H}_2\text{O} \longrightarrow \text{CH}_3\text{OH} + \text{NaOH} \tag{7}$$

$$\text{NaOH} + \text{CO} \longrightarrow \text{HCOONa} \tag{8}$$

$$\text{CH}_3\text{ONa} + \text{CO}_2 \longrightarrow \text{CH}_3\text{OCOONa} \tag{9}$$

### 3.1.4. Improvements

Improvements were achieved by (1) increasing temperatures and CO partial pressure, (2) utilization of alternative catalysts and (3) adding of co-catalysts or auxiliary materials to improve the resistance of the process against $CO_2$ and water.

High pressure processes operate with increased CO partial pressures between 9 and 180 MPa and higher temperatures. This improve the space-time-yield (STY), but reactors are required with significantly higher investment costs. Auer et al. describe a STY of 900 g $L^{-1}$ $h^{-1}$ at 100 °C and 15 MPa in comparison to 800 g $L^{-1}$ $h^{-1}$ of the established process. This corresponds with a methanol conversion of 57% which is an almost twofold increase [13]. In order to improve the current process investigations regarding new catalysts, both homogeneous and heterogeneous, have been carried out. As homogeneous catalysts basic catalysts and transition metal complexes have been reported. The latter are more resistant towards water and $CO_2$ but are expensive and need drastic conditions. For heterogeneous catalysis basic resins are eligible [10].

Homogeneous Catalysts

Next to sodium methoxide, potassium and lithium methoxide have been tested as potential homogenous catalysts. It was found that the rate of carbonylation correlates indirectly with the ionization potentials of the alkali metals. A low ionization potential results in an increased electron density on the O atom in the methoxide, which is essential for the nucleophilic attack on the CO (Equation (2)). Potassium, having the lowest ionization potential of 4.32 V, followed by sodium (5.12 V) and lithium (5.36 V), showed the fastest reaction rate over sodium methoxide and lithium methoxide ($r(KOCH_3) > r(NaOCH_3) > r(LiOCH_3)$) [14].

The presence for alkali formate seems to improve the catalytical effect of the methoxide. Especially the system potassium methoxide (0.8 wt.%) and potassium formate (5 wt.%) improves the MF production by 27% compared to the reaction without formate addition (5.5 MPa, 85 °C). The utilization of sodium and rubidium formate works, too, but yields lower MF concentrations (Na formate 13.0 wt.%, Rb formate 13.6% and K formate 15.2%) [15].

Many studies aimed at reducing water and $CO_2$ sensitivity by the addition of auxiliary materials to Na or K methoxide. Resistance towards $CO_2$ can be improved by adding tertiary alkyl amines and terminal epoxides to the sodium methoxide catalyst. In the

presence of $CO_2$, MF yield and methanol conversion reduce drastically to 6% and 8% after 17 h. Addition of 20 mmol triethylamine ($NEt_3$) and 150 mmol epoxy butane (EpBu) to sodium methoxide results in a methanol conversion of 82% and a MF yield of 64%. The lower MF selectivity in comparison to the reaction without the addition can be explained by a formed ether-alcohol species [16]. A reduction of catalyst consumption was achieved by adding sodium or potassium oxaperfluoroalkane sulfonate and a strong organic base ($pK_a > 8.7$; e.g., guanidine) to the system [17].

Further investigations have shown that the addition of a group 6 and/or 8 metal carbonyl to the sodium methoxide catalyst enhances MF production and reduces sensitivity towards water impurities. The rate limiting step for MF formation via carbonylation is the attack of methoxide on CO. The addition of group 6 metal carbonyls increases the activity of the catalyst significantly. The metal carbonyl is characterized by a higher electrophilicity in comparison to free CO, which facilitates the nucleophilic attack of the methoxide. The methoxide anion activates the metal coordinated CO to form a short-lived methoxycarbonyl (metalloester) complex (Equation (10)) [4].

$$\left(OC\right)_{n-1}\!M\!-\!CO \;+\; H_3C\!-\!O^- \;\rightleftharpoons\; \left[\left(OC\right)_{n-1}\!M\!-\!\underset{\underset{O}{\|}}{C}\!-\!OCH_3\right]^- \tag{10}$$

M = Ni, W, Fe, Ru, Os, Cr, Pd, Pt

At temperatures $\geq 100\ °C$, the metalloester is protonated by an alcohol to eliminate MF and regenerate methoxide. Furthermore, the metal carbonyl is regenerated under CO atmosphere [4]. Metal carbonyls of Cr, Mo, W and Ni are known to enhance methanol conversion under mild conditions. In the absence of methoxide only low concentrations of MF are detectable. The presence of methoxide increases the catalytic activity of these carbonyls and a significant amount of MF is produced, except for $Ni(CO)_4$ which produces only methanol. Tungsten carbonyl $W(CO)_6$ is known to sextuple the catalyst activity compared to potassium methoxide alone [18,19]. Addition of $Mo(CO)_6$ to methoxide increases MF production with CO and syngas present, but less significant than $W(CO)_6$. The study of Jali et al. shows that an excess of methanol enhances MF production, through regenerating the catalyst (Figure 2) [4].

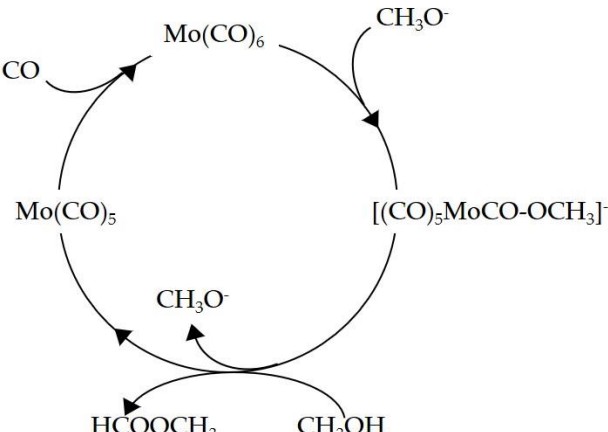

**Figure 2.** Catalysis mechanism of methanol carbonylation with methoxide and $Mo(CO)_6$ for production of MF [4].

Alternative homogeneous catalysts for MF formation by carbonylation are nanocluster catalyst of Cu, Ru, Pt, Pd, Rh, Ir and Au. The highest CO conversion of 32.7% was observed with Cu at 100 °C and 1.0 MPa CO in 2 h, corresponding to an activity of 2.7 $mol_{MF}$ $mol_{Cu}^{-1}\ h^{-1}$. The MF selectivity was 100%. The advantage of this catalyst lies in $CO_2$ not

affecting the carbonylation reaction. Furthermore, low water concentrations (<10 mol% in methanol) did not take effect on both activity and MF selectivity [20]. While the exact reaction mechanism is still unknown, there are several hypotheses which may serve to explain the processes on a molecular level. For instance, the working group of He et al. proposed that:

(1) Adsorption and oxidation of methanol on Cu sites forms $Cu-OCH_3$ and Cu-H species;
(2) CO adsorption and its insertion in the Cu-O bond in $Cu-OCH_3$ forms $CH_3OC=O$ intermediates;
(3) Reductive elimination of the $CH_3OC=O$ intermediates provides MF [20].

Besides Cu, nanocluster catalysts Ru, Pt and Rh were suitable for MF formation. The activity decreases to 1.3, 0.8 and 0.7 $mol_{MF}\ mol_{metal}^{-1}\ h^{-1}$ on Ru, Pt and Rh (vs. Cu 2.7 $mol_{MF}\ mol_{metal}^{-1}\ h^{-1}$). Ir, Au and Pd show a very low activity but just like the other catalysts a high MF selectivity (nearly 100%) [20].

Additionally, iron clusters of type $[Et_4N][Fe_3(CO)_9E]$ with E = S, Se or Te are suited for MF production. The reaction rate is ten times higher compared to sodium methoxide as a catalyst. In comparison to the reaction with methoxide the reaction rate is affected by CO pressure, but not by methoxide concentration [21].

Heterogeneous Catalysts

Promising heterogeneous catalysts are basic resins. Di Girolamo et al. compared the resins Amberlyst A26, IRA 910 and IRA 400 with the homogeneous catalyst methanolate at 5 MPa. Amberlyst A26 is very active at temperatures lower 60 °C. After 5 h a methanol conversion of 91% and a TOF of 78 $h^{-1}$ could be achieved. According to the manufacturer information at temperatures higher 60 °C starts the degradation of the resin. However, Di Girolamo et al. could observe catalytic activity at 67 °C (83% methanol conversion, TOF = 73 $h^{-1}$). The Amberlite IRA 400 is a gel resin with the same functional group (Ⓟ-$CH_2N(CH_3)_3^+Cl^-$), like Amberlyst A26, and shows quite analogous result to Amberlyst A26 (methanol conversion 91%, TOF = 73 $h^{-1}$) for temperature up to 67 °C. Under comparable conditions carbonylation of methanol with sodium methanolate achieved only a methanol conversion of 55% at 57 °C and 78% at 67 °C and thus, lower than with the tested resins. However, the macromolecular resin Amberlite A910 with the functional group Ⓟ-$CH_2N(CH_3)_2(CH_2CH_2OH)^+Cl^-$ displayed a very poor catalytic performance (methanol conversion < 26%, TOF < 11). Beside the low formation of DME (Equation (4)), an inactivation of the catalytic activity through hydrolysis reactions (Equations (11)–(13)) could be observed [10].

$$\text{Ⓟ-}CH_2N(CH_3)_3{}^+CH_3O^- + H_2O \longrightarrow CH_3OH + \text{Ⓟ-}CH_2N(CH_3)_3{}^+OH^- \qquad (11)$$

$$\text{Ⓟ-}CH_2N(CH_3)_3{}^+OH^- + HCOOCH_3 \longrightarrow \text{Ⓟ-}CH_2N(CH_3)_3{}^+HCOO^- + CH_3OH \qquad (12)$$

$$\text{Ⓟ-}CH_2N(CH_3)_3{}^+CH_3O^- + CO \longrightarrow \text{Ⓟ-}CH_2N(CH_3)_3{}^+HCOO^- \qquad (13)$$

The advantage of these resins is the possibility to regenerate this with caustic washing [10]. In comparable investigations, Iwase et al. showed that it is possible to reuse the resin more than four times without significant changing of the activity [22].

Vapor-gas methanol carbonylation could be conducted by the utilization of nanoscaled platinum group metal catalyst. Xu et al. used a palladium catalyst (mixed with Cu, Co, Fe and fixed on activated carbon) in a fixed bed-reactor at 100 °C and 1.5 MPa. It could be measured a MF selectivity of 97%, a CO conversion of 67% and a space-time yield of 998 $gL^{-1}h^{-1}$. Comparable results could be achieved by utilization of other carrier materials (e.g., zirconia, titanium dioxide, silica and zinc oxide) and rhodium or iridium instead of palladium [9].

### 3.2. Dehydrogenation of Methanol

3.2.1. Process

Methanol dehydrogenation to MF is an endothermic process in which hydrogen is generated as a side-product (Equation (14)). Thus, the produced hydrogen can be reused to form methanol by $CO_2$ hydrogenation.

$$2\,CH_3OH \longrightarrow HCOOCH_3 + 2\,H_2 \qquad \Delta H = +98.9\ kJ\ mol^{-1} \qquad (14)$$

The first patent describing this process was submitted 1925 by US Industrial Alcohol Co, the first industrial methanol dehydrogenation was achieved by Japan Mitsubishi Gas Chemical Co. Ltd. by using $Cu\text{-}Zn\text{-}Zr/Al_2O_3$ catalyst. The reaction is carried out at 250 °C, ambient pressure and gain a methanol conversion of 58.5% as well as a selectivity of MF of 90% [1].

Dehydrogenation of methanol for MF production is classically performed in a fixed-bed-reactor with a copper-based catalyst. Many studies have shown that copper is the active species for the reaction [23–26].

Despite the low MF concentration, the corrosive properties lead to the necessity of titanium columns for distillative separation, whereby the production cost increase drastically [5].

The methanol dehydrogenation is used in small-scale industrial production, but is limited due to the thermodynamic equilibrium [1].

3.2.2. Mechanism

The mechanism of dehydration of alcohols to aldehydes and subsequent formation of esters has been investigated in recent studies (2017). It could be shown that both basic and slightly acidic centers are essential. In the first step, the methanol binds onto copper atoms situated onto the surface and forms alkoxide. The proton is delivered to a base site and is formed via β-hydride elimination to an aldehyde and subsequently to molecular hydrogen. The aldehyde coordinated with the non-coordinated electron pair of the oxygen on the acidic center. This activates aldehyde promotes the condensation with alcohols to furnish hemiacetal. Finally, the hemiacetal is dehydrogenated by the adjacent surface base and metallic Cu to give the final product (Figure 3) [27].

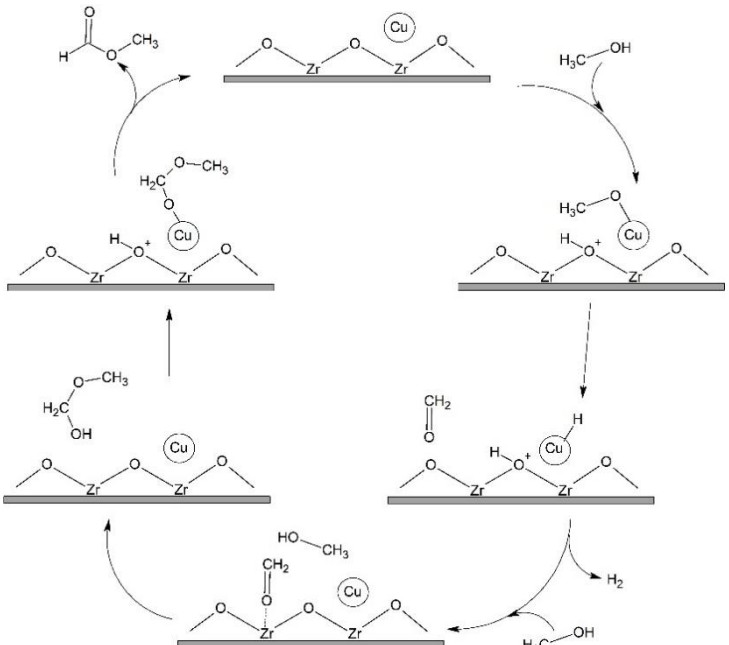

**Figure 3.** Proposed reaction mechanism for methanol dehydrogenation to MF [27].

Various studies showed in batch reactor experiments that no hemiacetal occurs in the reaction products of dehydration. In these studies, the alcohol was dissolved in mesitylene, and the dehydrogenation was carried out on differently doped heterogeneous catalysts at 170 °C. The fact that no hemiacetal was measured as a reaction product suggests that the formation of hemiacetal is the rate limiting step. Another mechanism, the dimerization of two molecules of aldehyde to form an ester, could be not observed under these conditions [27].

### 3.2.3. By-Products

During the methanol dehydrogenation process, dimethyl ether (DME) is usually produced as a by-product via the acid-catalyzed methanol dehydration reaction. The acidic centers are located on the carrier material, with shows the influence of it [24]. In addition, the released water can easily react with the intermediate formaldehyde, resulting in the formation of $CO_2$ (Equations (15) and (16)) [28].

$$2\,CH_3OH \longrightarrow CH_3OCH_3 + H_2O \qquad \Delta H = -23.4\;kJ\;mol^{-1} \qquad (15)$$

$$CH_2O + H_2O \longrightarrow CO_2 + 2\,H_2 \qquad \Delta H = -78.5\;kJ\;mol^{-1} \qquad (16)$$

At higher temperature (decomposition temperature: 85 °C) decomposition of MF into CO takes places [24,25].

### 3.2.4. Catalyst

As the exact process conditions as well as the catalysts composition is unreleased until today, a high interest of research remains. Typical investigated catalysts are $Cu/SiO_2$, $Cu/Cr_2O_3$, $Cu/ZrO_2$, $Cu/ZnO$, $Cu/MgO$ and $CuAl_2O_4$ [1]. Both preparation method and carrier material have a significant influence on the catalyst activity. Furthermore, copper concentration as well as addition of further elements affect the catalytic process [1]. Both preparation method and carrier material have a significant influence on the catalyst activity. Furthermore, copper concentration as well as addition of further elements affect the catalytic process.

Influence of the Preparation Method

Guerreiro et al. synthesized $Cu/SiO_2$ catalyst via ion exchange and subsequently characterized them. They observed that the methanol conversion increase with rising copper loading and reach a maximum of 55% at 3 wt.% Cu at 240 °C. The MF selectivity, however, is independent of the Cu loading (60–65%) [29]. A similar catalyst, prepared by wet-impregnation, reached a higher MF selectivity (80%) at a comparable methanol conversion (55%). The Cu load is slightly lower (2.6 wt.%) but the specific surface area is much higher (390 vs. 116 $m^2\;g^{-1}$) [30]. A much higher surface area (313.4 $m^2\;g^{-1}$) could be achieved by a sol-gel technique. These $CuO\text{-}SiO_2$ co-gels produce MF with high selectivity (<80.8%) but at low methanol conversion (>35.46%) and higher copper content (9.1 wt.%) [25]. Sodesawa et al. observed, that $Cu/SiO_2$ catalyst prepared by ion exchange method show a longer lifetime as compared with those of Cu catalysts prepared by other methods. It was observed that the TOF increases with growing Cu dispersity. It was concluded that the highly dispersed Cu particles on the $SiO_2$ support cannot be easily aggregate during the reaction [31,32]. The $Cu\text{-}SiO_2$ catalyst prepared by ion exchange exhibited the largest influence of the reaction temperature and gave the highest MF yield in the range of 180 to 300 °C. TPR measurements shows a shift of the reduction peak in dependence of the preparation method (ion exchange: 190 °C, impregnation: 200 °C and precipitation: 230 °C), thus the catalyst prepared by ion exchange is more easily reduced by $H_2$ than the other catalysts [33].

Influence of Copper Loading

Besides preparation method and carrier material, copper content has an influence on MF production. Tonner et al. varied the Cu loading of copper chromite and Cu onto SiO$_2$. Regardless of the carrier material the methanol conversion as well as the MF yield increase with rising copper load. The MF selectivity is nearly independent from the Cu load [34]. Analogous observations were conducted by Gao et al. with zeolite-supported Cu-catalysts (Figure 4) [28].

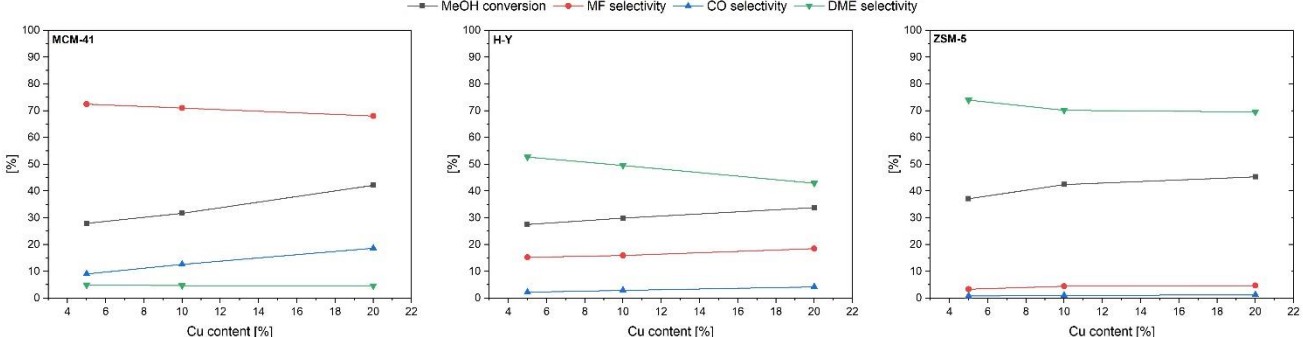

**Figure 4.** Influence of the Cu-content of different zeolite-based Cu-catalysts on methanol conversion and selectivity of MF, CO and DME [28].

Influence of Carrier Material

The carrier material also was shown to exert a significant effect on MF selectivity as well as TOF (Table 2). Here, too, it can be seen, that an increasing Cu dispersity improves the TOF of the catalyst, at least at low temperatures. At higher temperature, the opposite trend could be observed. Independent of the carrier material decrease the MF selectivity with increasing temperature due MF decomposition. At both tested temperatures, the highest MF selectivity could be observed with TiO$_2$ as a carrier material, although it showed the least dispersity. A reason for this could be strong metal-support interactions because of Cu catalyzed reduction of TiO$_2$ to a lower oxide [32].

**Table 2.** Influence of carrier material on MF selectivity, TOF and Cu dispersity depending on reaction temperature [32].

| | Dispersity [%] | Selectivity MF [%] | | TOF [s$^{-1}$] | |
|---|---|---|---|---|---|
| | | 190 °C | 250 °C | 190 °C | 250 °C |
| 1.5 wt.% Cu/SiO$_2$ | 75.5 | 83 | 25 | 0.033 | 0.086 |
| 1.5 wt.% Cu/ZrO$_2$ | 60.7 | 83 | 63 | 0.018 | 0.10 |
| 1.5 wt.% Cu/TiO$_2$ | 23.0 | 92 | 91 | 0.0077 | 0.18 |

Cu/ZnO-based catalysts are known to produce MF via dehydrogenation of methanol. However, several processes lead to inhibition of the reaction. Firstly, the stability depends on the composition of the reactants. Furthermore, decomposition of formed formaldehyde onto the catalyst inhibits dehydration. Moreover, reduction of ZnO leads to formation of alloy with Cu and thus reduction of catalytic activity [35–38]. Experimental data show that CO and methanol reduce ZnO in Cu/ZnO, while H$_2$ does not when below 500 °C. The prerequisite is the presence of copper metal. The reduction of ZnO results in a decreasing surface area due to alloy formation and thus reduction of the catalytic activity [37]. During methanol dehydration CO is formed by water-gas-shift-reaction from CO$_2$, which in turn is a decomposition product from formaldehyde (Equation (16)). Furthermore, polymerization of formaldehyde on the catalyst surface deactivates the catalyst [36].

An improvement of the stability of the Cu/ZnO-catalyst can be reached by incorporation of Al or Cr. In this way, reduction of ZnO reduction could be suppressed, whereby

catalytic activity maintained. Jung et al. could be shown that ZnO in Cu/ZnO-catalyst reduced in a range of 200 to 400 °C, the temperature range of methanol dehydration. By adding Al to the catalyst, the reduction can be suppressed up to 330 °C and by adding Cr even up to 500 °C. These results are reflected in the methanol conversions. Cu/ZnO catalyst reached a conversion of only 10%, while CuZnAl and CuZnCr obtained 30% and 40%. The MF selectivity could be only improved be incorporation of Al (80% vs. 82%). Methanol dehydration with a CuZnCr catalyst shows only a MF selectivity of 55%, which the authors explained by MF decomposition onto the catalyst into CO [35].

Very good results (methanol conversion: 48% and MF selectivity: 99%) were achieved with a $Cu/Cr_2O_3$ catalyst at 207 °C and atmospheric pressure [1]. However, US Air Products and Chemicals Inc. adopted this catalyst at 0.17–0.68 MPa and 170–210 °C and realized high selectivity to MF (>98%) but at the price of lower methanol conversion rates (13% per pass) [1].

Sato et al. produced an effective $CuO/Al_2O_3$ catalyst by amorphous citrate process. The highest MF formation activity could be observed at a Cu-Al-ratio of 1:2, the results are shown in Table 3. An increasing temperature improves the methanol conversion, but the MF selectivity decrease due MF decomposition. Furthermore, the catalytic activity depends on the calcination temperature of the sample. The maximum MF formation rate increase with increasing calcination temperature and the reaction temperature, at which the maximum rate is reached, rise. For example, of the catalyst calcined at 600 °C, the maximum MF formation rate was attained at 250 °C and the MF selectivity was only 33.8%. On the other hand, the maximum MF formation rate of Cu-Al-catalyst calcined at 1100 °C was reached at 320 °C and a quantitative MF selectivity was determined. Characterization of the catalysts via XRD and TPR showed that samples calcined below 1000 °C are partially reduced during methanol dehydration, and the selectivity to MF decrease with increasing Cu(0) content. Although Cu(0) is active and selective at temperatures below 210 °C, Cu(0) decomposes MF to CO above 210 °C. Samples calcined at 1100 °C are not reduced at 310 °C, which is an indication that Cu(II) species in the $CuAl_2O_4$ plays an important role as active sites at temperatures above 250 °C and Cu(I) species up to 290 °C [39].

**Table 3.** Production of MF catalyzed by $CuO/Al_2O_3$ (Cu:Al 1:2) [39].

| Reaction Temperature [°C] | Methanol Conversion [%] | Selectivity MF [%] | Selectivity CO [%] | Selectivity $CO_2$ [%] | Selectivity DME [%] |
|---|---|---|---|---|---|
| 270 | 26.2 | 95.4 | 4.6 | 0 | 0 |
| 310 | 63.6 | 57.6 | 15.9 | 6.8 | 20.3 |

An improvement of the catalytic performance could be reach by using MgO as a carrier material. CuMgO catalyst shows a methanol conversion in a range of 11.7 to 16.7% and MF selectivity from 62.5 to 88.1%. Analogous to copper catalyst onto other carrier, the methanol conversion as well as a MF formation rate increase with rising copper content. However, an excess amount of Cu harms the reaction. The reason for this is the formation of Cu cluster at higher Cu loading, whereas methanol dehydration at lower Cu concentration favors the formation of nanoparticles. This is also reflected in the fact that the lifetime of the catalyst, up to a Cu-Mg ratio of one, increases with increasing copper content. Higher copper-containing catalysts, such as $Cu_5MgO_5$ and $Cu_7MgO_3$, also displayed a higher CO selectivity compared to those with lower Cu content. The methanol conversion decreases with increasing reaction time due to coke formation; however, the MF selectivity remains constant. An excess of MgO offers more basic sites, with results in deactivation of the catalyst due to polymerization reactions and coke formation A prevention of deactivation and enhancement of stability could be achieved by adding palladium to CuMgO due to inhibition of the basic sites. Moreover, palladium decrease the reducibility of the catalyst and thus reduce the selectivity towards CO and $CO_2$ and increase the MF selectivity. Addition of Pd to $Cu_3MgO_7$ enhance the methanol conversion

from 6.1 to 15%. The supplementation of $Cu_5MgO_5$ improves the methanol conversion and MF selectivity slightly but improves the lifetime significantly. Long-term tests showed that $Pd/Cu_5MgO_5$ can maintain 80% of the initial activity after 100 h reaction, whereas the activity of the catalyst without Pd drop down to approximate 40%. Furthermore, it could be shown, that $Cu_5MgO_5$ could be used at least four times without strong effect on methanol conversion and MF selectivity [24].

Besides classical carrier materials such as silica, clays are suitable for MF production. Matsuda et al. compared different copper-exchanges clays concerning methanol dehydration. Cu-laponite and Cu-TSM (fluoro tetrasilicic mica) are able to produce MF with high selectivity (82.2% and 89.3%) at a methanol conversion of 35.8% and 16.7%. Cu-saponite and Cu-montmorillonite predominantly catalyzed DME formation due to their acidity. The activity of Cu-laponite increase with rising pretreatment temperature, which is explainable by water desorption. During dehydration methanol conversion could be improved by increasing reaction temperature. However, with rising temperature the MF selectivity decrease due MF decomposition into CO and $H_2$. In the case of Cu-TSM decrease, the activity will have a rising treatment temperature, because of decreasing interlayer distance [40].

The application of the acidic carrier $Al_2O_3$ promotes, particularly, the formation of DME and formaldehyde. With decreasing acidity, the MF selectivity rises significantly due methanol conversion and DME selectivity decline [28]. Less acidic and especially amphoteric oxides such as $SiO_2$, $Ga_2O_3$, $In_2O_3$, $Cr_2O_3$ and ZnO form predominantly MF as a product. $ZrO_2$ can be used as a carrier, but it also supports the decomposition of MF to CO. Basic carriers, such as MgO are able to produce MF, but they are also decomposing methanol resulting in low MF selectivity and yield. With rising basicity selectivity for CO increased due to decrease MF formation.

Addition of Further Elements

US patent 5144062 described the production of a $Cu/Cr_2O_3$ catalyst with and without addition of sodium (Figure 5). Due to addition of sodium (0.2 wt.%) to the catalyst methanol conversion as well as MF selectivity and yield increase. This effect was observed especially at lower Cu loading. Without sodium addition, methanol conversion and MF selectivity could be improved by increasing Cu content. The MF yield is only slightly affected. This observation is changing by adding sodium to the $Cu/Cr_2O_3$ catalyst. In this case, an increasing Cu loading reduce both methanol conversion and MF yield. The MF selectivity could be improved slightly [41].

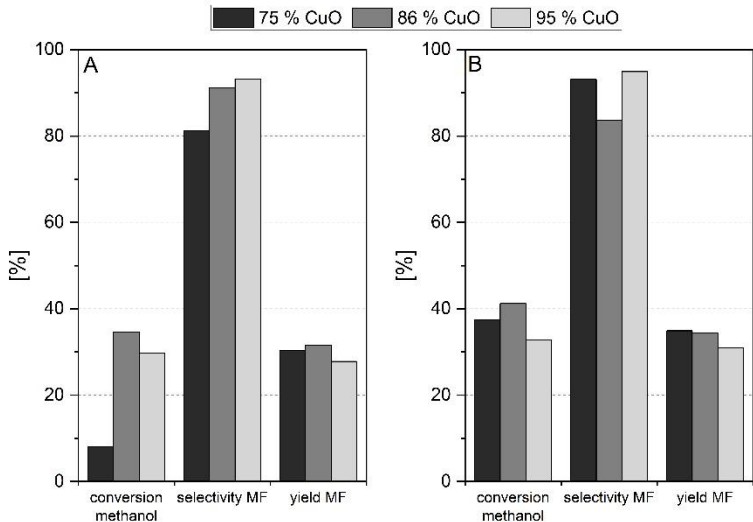

**Figure 5.** Influence of CuO content of $CuO/Cr_2O_3$ catalyst without (**A**) and with (**B**) sodium for dehydrogenation of methanol at 180 °C and 0.1 MPa [41].

Different patents describe the production of catalysts comprising Cu, Zn, Zr and Al. They are able to produce MF with high yield (<49.1%) and high selectivity (<92.7%) at comparable high methanol conversion (>61.5%) [42,43].

Moreover, methanol dehydrogenation with Cu-Zr-K catalyst at 0.1–2.0 MPa and 200–500 °C produce MF with high selectivity of 90% at low methanol conversion (32%) [1]. Similar observations were made with Cu-$B_2O_3$/$SiO_2$ catalyst. At 270 °C and ambient pressure MF selectivity of 85–91% and methanol conversion of 22–25% was achieved [1].

Rodriquez-Ramos et al. investigated the methanol dehydration with copper-containing perovskite-type oxides $LaM_{1-x}Cu_xO_3$ with x in the range of 0.2 to 0.6. All tested oxides catalyzed MF production with a selectivity of approximate 65% regardless of Cu content. The MF yield was determined in the range from 10 to 29%. Unmodified $LaMnO_3$ produced no MF. Furthermore, unmodified $LaTiO_3$ are not able to catalyze MF formation. A modification with copper results in an MF yield of approximate 25% regardless of Cu content. The selectivity decreases with decreasing Cu load from 73% (x = 0.8) to 63% (x = 0.3) [44].

Cu-Free Catalysts

Although the focus of current investigations is on copper-containing catalysts, other catalysts are also suitable for dehydrogenation. For example, Pd/ZnO catalyst is highly selective (80%) for methanol dehydrogenation (methanol conversion 20.5%). Other palladium catalysts, such as Pd/$SiO_2$, Pd/$Al_2O_3$, Pd/MgO and Pd/$Cr_2O_3$ show no activity. Both palladium and ZnO alone show no catalytic activity, thus the synergetic effect of Pd and ZnO is necessary for MF production. Methanol dehydrogenation with Pd/ZnO prepared by precipitation is faster than that of Pd/ZnO prepared by wet-impregnation. Analogous to copper catalyst, the MF selectivity decrease with increasing residence time by formation of by-products [45]. In contrast to Cu/ZnO catalysts, the MF formation increase slightly with increasing reaction time. No deactivation was observed within the first 250 min. The underlying reaction mechanism based on formation of PdZn alloy due to reduction of ZnO. The formed intermediate formaldehyde adsorbed and stabilized on the alloy surface and transformed into MF [46].

3.2.5. Reactors for Methanol Dehydrogenation

The classical approach for dehydrogenation is the utilization of a fixed-bed-reactor, with gaseous methanol streaming through a catalyst bed. An alternative approach is the reaction in a slurry reactor. Yamakawa et al. compared these two approaches using the example of three copper catalysts. The gas-phase reaction reached only a methanol conversion of maximum 3% [47].

Guo et al. also used ion exchange to synthesize a Cu/$SiO_2$/ceramic composite. They compared the methanol dehydrogenation in a catalytic membrane reactor (CMR) with the reaction in a fixed bed reactor (FBR). The advantage of CMR is the removal of the formed hydrogen from the reactive zone by a permselective membrane and thus, shift the equilibrium towards the product MF. On this way a higher methanol conversion (57.3% vs. 43.1%) and a higher MF yield (50% vs. 36.9%) at constant MF selectivity (approximately 88%) in comparison to FBR could be observed [48].

3.2.6. Coupling of Dehydrogenation and Hydrogenation Reaction

During methanol dehydrogenation large amounts of hydrogen are released but unfortunately, the released hydrogen cannot be effectively used in a single methanol dehydrogenation process. It is possible, however, to couple the dehydrogenation with an exothermic hydrogenation reaction. For example, Gao et al. produce γ-butyrolactone (GBL) through coupling methanol dehydrogenation with hydrogenation of maleic anhydride (MA) over copper-based catalysts. The hydrogenation of MA is a strong exothermic reaction ($\Delta H = -221.06$ kJ mol$^{-1}$) and difficult to control, resulting in a low selectivity of the desired product GBL. The reaction includes three steps: (1) conversion of MA into dimethyl

maleate through esterification reaction (Equation (17)), (2) hydrogenation to form dimethyl succinate (DMS) (Equation (18)) and (3) production of GBL (Equation (19)) [28].

$$\text{(17)}$$

$$\text{(18)}$$

$$\text{(19)}$$

Gao et al. investigated different Cu/zeolite catalysts to coupling methanol dehydrogenation and MA hydrogenation. They observed that the catalytic activities of the tested catalysts were in order Cu/MCM-41 > Cu/Y > Cu/ZSM-5 > Cu/β, the inverse of their acidities. With increasing temperature (240–300 °C) the conversion of methanol (<53.1%) and MA (<99.5%) increase. The selectivity of GBL rise up to 71.2%, whereas the selectivity of MF decreases up to 63%. It could be also shown that the hydrogen formed during methanol dehydrogenation is sufficient for GBL production. Reaction heat from exothermic MA hydrogenation and endothermic methanol dehydrogenation reactions is compensated to some extent [28]. Unfortunately, no remarks regarding product separation were given. However, due the different boiling points (MF 32 °C, GBL 205 °C) a separation through distillation should be possible.

### 3.3. Oxidation of Methanol

The partial oxidation of methanol to form methyl formate is a strong exothermic process, which is thermodynamically more favorable than dehydrogenation (Equation (20)). Three groups of catalysts are distinguished, namely (1) noble metal and noble metal composites, (2) transition metal oxide composites and (3) TiO$_2$-based catalysts. The oxygen required for the reaction can be taken from water electrolysis, which is carried out to produce the hydrogen required for methanol synthesis. This, in turn, allows a holistic utilization of the valuable resource water (circular economy).

$$2CH_3OH + O_2 \longrightarrow HCOOCH_3 + 2H_2O \qquad \Delta H = -428.9 \text{ kJ mol}^{-1} \qquad \text{(20)}$$

Methanol oxidation to MF on metal oxide catalysts involves rate-determining C-H activation steps to form formaldehyde. Subsequent formaldehyde reacts with intermediates derived from methanol or formaldehyde to MF [1].

There is a diverse range of different catalytic systems, active in partial methanol oxidation, that are tested in lab scale. Considerations of usage in industrial scale have been made for less sophisticated systems and some systems are patented [49,50]. To the authors best knowledge, none of the presented catalytic systems have been tested in a pilot plant scale study so fare [51,52].

Catalysts

Most commonly, TiO$_2$-based catalysts are examined, though research interests have somewhat shifted from research of V$_2$O$_5$-TiO$_2$ catalysts towards photooxidation over TiO$_2$ and TiO$_2$ heterojunctions [53].

Forzatti et al. were the first to report selective gas phase oxidation of methanol to methyl formate over a coprecipitated $V_2O_5$-$TiO_2$ catalysts at temperatures below 200 °C. When utilizing a catalyst with a low V/Ti atomic ratio, methanol conversion as well as MF selectivity increase with rising temperature, whereby MF selectivity goes through a maximum. At 170 °C, methanol conversion rates of approximately 80% can be achieved with a MF selectivity of approximately 78% [52]. In a comparative FTIR study of methanol adsorption and oxidative reaction on the catalyst, Feil et al. explained the methyl formate/formaldehyde selectivity with the chemisorption bond of initial methoxy species on the catalyst surface. The proposed reaction mechanism involved the following steps: (1) the rate-determining chemisorption of methanol, (2) the surface reaction of methoxy species to oxymethylene species and (3) the reaction of both to form methyl formate [51]. In a comparative FTIR study of methanol adsorption and oxidative re-action on the catalyst, Feil et al. explained the methyl formate/formaldehyde selectivity with the chemisorption bond of initial methoxy species on the catalyst surface and proposed a reaction mechanism, involving the rate-determining chemisorption of methanol, the surface reaction of methoxy species to oxymethylene species and the reaction of both to form methyl formate [51].

In their own FTIR-study, Busca, Elmi and Forzatti refined and expanded the proposed mechanism by a Canizarro-type disproportionation. Moreover, they compared V-Ti-O catalysts prepared by coprecipitation and impregnation and found that methanol conversion and MF selectivity goes through a maximum at 10 K higher for the co-precipitated catalyst [54]. The authors considered the application of the catalyst system in a methyl formate/formaldehyde co-production process, requiring a compromise between methyl formate production revenues and formaldehyde separation costs [55].

More recently, Kaichev et al. have presented an improved $V_2O_5$-$TiO_2$ catalyst design, achieved by washing the impregnated catalyst with nitric acid prior to calcination. This approach ensures the formation of a dispersed vanadium monolayer of polymeric VOx species on the support, rather than $V_2O_5$ crystallites [56]. In this way, MF selectivity can be pushed to 90% at 150 °C and 50% methanol conversion. At temperatures below 120 °C, selectivity of the oxidation shifts towards dimethoxymethane. The formation of polymeric VOx species on the support is essential in insuring high partial oxidative performance of the catalyst. For catalysts with supports, that do not promote formation of these species, MF selectivity decreases drastically [57].

Even further improvement was achieved by Liu et al. by co-precipitation of vanadium oxide and titanium sulfates. The acquired vanadia-titania-sulfate catalyst achieves nearly complete methanol oxidation to MF at 145 °C with a methanol conversion of 98.7% and MF selectivity of 98.6%. Sulfate acts as a promoter of catalytic activity by lowering the energy barrier oxidative dehydrogenation of chemisorbed MeOH species [58,59].

A nearly quantitative conversion of methanol to MF was achieved by Zhang et al. by using graphene confined nano-oxides. This catalyst system consists of VTiO nanoparticles, which are located in a graphene cage. This system allows the production of MF at low-temperatures (135 °C) with methanol conversion rates of 98.8%. Both experiments and simulations explained the high conversion rates with shell/core interfacial electronic structure and the surface chemistry of the catalysts. The catalyst's stability over 500 h is impressive [60].

In the last 15 years, a second group of catalysts has been extensively tested on their ability to oxidate hydrocarbons. These are unsupported and supported noble metal catalysts often with a nanoscale structure. Considerations of utilization of the materials, especially concerning gold nanocomposites and its possible catalytic activity in oxidation, were first formulated by Haruta in 2007 [61].

In a remarkable publication, Wittstock et al. demonstrated the high catalytic activity of unsupported, nanoporous gold in the gas phase oxidation of methanol to MF at temperatures below 100 °C and ambient pressure. The catalyst was obtained by dealloying a silver-gold alloy with nitric acid to form a nanoporous gold structure [50]. At room temperature 10% of methanol in the gas phase was converted exclusively to MF. At elevated

temperatures of 80 °C, 60% methanol conversion was achieved with a MF selectivity of 97%. TOF was measured at 0.26 s$^{-1}$, with pore diffusion being the rate determining step. The catalyst does retain its activity for 7 days on stream at 30 °C. At 60 °C, a slight, but reversible activity decreases of 6% per day was observed. Most notably, the research showed the effect of residual Ag content on the selectivity of partial oxidation. With increasing Ag content, the selectivity towards MF oxidation decreases. At 10% atomic Ag, MF was not produced any more [61].

This observation is truly important, because it demonstrates the ability to tune for a specific selectivity of partial oxidation by varying the bimetallic content of nanoscale alloy composition catalysts. In consequence, a number of various bimetallic noble metal catalysts have been tested (Table 4).

**Table 4.** Noble metal composite catalysts for the partial oxidation methanol to MF.

| Noble Metal Composition | Support | Temperature [°C] | Methanol Conversion [%] | Selectivity MF [%] | Reference |
|---|---|---|---|---|---|
| Au1.0-Ag0.2 | Al-fiber | 170 | 42.0 | 82.0 | [62] |
| Au1.0-Ag1.0 | TiO$_2$ | 35; UV | 82.5 | 87.5 | [63] |
| Au2.0-Pd1.0 | Graphene | 70 | 90.2 | 100.0 | [64] |
| Au2.0-Pd1.0 | SiO$_2$ | 130 | 57.0 | 72.7 | [65] |
| Au1-0-Pd1.0 | TiO$_2$ | 30 | 15.0 | 70.0 | [66] |
| Au0.5-Pd0.5 | TiO$_2$ | 30; UV | 85.0 | 70.0 | [67] |
| Pd0.65-Pt0.35 | TiO$_2$ | 50 | 78.0 | 67.0 | [68] |
| Pd1.0-Cu1.0 | SiO$_2$ | 30; UV | 53.0 | ~80.0 | [69] |
| Ag2.0-Pt0.5 | SiO$_2$ | 100 | 99.5 | 58.9 | [70] |

In these catalysts, the nature of its nanostructure has a high impact on catalytic performance and changes in preparation protocol leads to differing catalytic performance, as demonstrated by Wojcieszak et al. in the case of supported Pd nanoparticles. Catalysts prepared by the microemulsion method show different catalytic properties depending on the used surfactant in preparation, since these have an influence on nanoparticle size. Smaller Pd nanoparticles are generally more active, but less selective towards MF. Methanol conversion of studied catalysts ranges from 20% to 78% with a MF selectivity range from 52% to 100% at 80 °C and ambient pressure [71]. The chosen support material does also affect the catalytic activity, since it influences the degree of oxidization of the active Pd species [72].

The last and smallest group of tested catalyst for the partial oxidation of methanol are supported transition metal oxide catalysts. Li et al. tested ZrO$_2$ supported RuO$_x$ catalysts and achieved 96% MF selectivity at methanol conversion rates of 15% at 100 °C and ambient pressure [73]. The performance was achieved at low Ru surface coverage and high amount of RuO$_4$ species, achieved by high temperature treatment of the catalyst [74]. Liu et al. showed the catalytic activity of an ReO$_x$ impregnated CeO$_2$ support and attained methanol conversion rates of up to 16 mmol g$_{cat}$$^{-1}$ h$^{-1}$ and MF selectivity of up to 88.5% at 240 °C and atmospheric pressure, depending on the catalysts Re loading [75].

### 3.4. Esterification of Methanol and Formic Acid

MF can be synthesized by esterification of methanol and formic acid with strong acid catalyst (Equation (21)) [1,76].

$$CH_3OH + HCOOH \longrightarrow HCOOCH_3 + H_2O \qquad \Delta H = -21.3 \text{ kJ mol}^{-1} \qquad (21)$$

The disadvantage of this process is the high consumption of formic acid and corrosion of equipment, which is why, with a few exceptions (especially in China), it has no industrial application [1].

### 3.5. Hydrogenation-Condensation with Methanol

Hydrocondensation of $CO_2$ with methanol is an exothermic reaction with allows the production of MF in an eco-friendly way (Equation (22)).

$$CH_3OH + CO_2 + H_2 \longrightarrow HCOOCH_3 + H_2O \qquad \Delta H = -25.4 \text{ kJ mol}^{-1} \qquad (22)$$

Most studies are based on homogeneous catalysts such as phosphine complexes of transition metals (e.g., $[RuHCl(PPh_3)_3]/BF_3$ and $RuCl_2(PPh_3)_3/DBU$) or carbonyl metalates of transition metals (e.g., $[HFe_3(CO)_{11}]^-$ and $[H_3Ru(CO)_{12}]^-$) [1,77,78]. Recent investigations target heterogeneous catalysts such as $Pd/Cu/ZnO$ and $Cu/ZnO$ onto $Al_2O_3$. Table 5 shows an overview of catalyst systems already investigated for the hydrocondensation of $CO_2$ with methanol.

**Table 5.** Different catalysts for hydrocondensation of $CO_2$ with methanol.

| | Catalyst | Reaction Condition | TON | MF Selectivity | Reference |
|---|---|---|---|---|---|
| homogeneous | $RuCl_2(PMe_3)_4$ | 80 °C, 64 h, 20 MPa | 3500 | 34% | [79] |
| | $[Ru(N\text{-}triphos^{Cy})(tmm)]$ | 60 °C, 18 h, 12 MPa | 9542 | 94% | [77] |
| | $[RuCl_2(dppe)_2]$ | 80 °C, 15.5 h, 13 MPa | 12,900 | n.d. | [80] |
| | $Fe(BF_4)_2/triphos\ 1$ | 100 °C, 20 h, 3 MPa | 292 | 56% | [81] |
| | $RuCl_2(PPh_3)_3/DBU$ | 140 °C, 40 h, 2 MPa | 1510 | 59% | [82] |
| | pDPPE | 160 °C, 12 h, 8 MPa | 3401 | n.d. | [83] |
| | $RuCl_2(PMe_2(CH_2)_2Si(OEt)_3)_3$ | 100 °C, 64 h, 13 MPa | 3180 | n.d. | [84] |
| heterogeneous | $Pd/Cu/ZnO/Al_2O_3$ | 140 °C, 0.5 h, 14 MPa | 109 | n.d. | [85,86] |
| | $Cu/ZnO/Al_2O_3$ | 140 °C, 0.5 h, 14 MPa | 131 | n.d. | [86] |
| | $Au/ZrO_2,\ Au/CeO_2,\ Au/TiO_2$ | 140 °C, 1 h, 16 MPa | 204 | >99.9% | [87] |
| | $Ag/SiO_2,\ Au/SiO_2,\ Cu/SiO_2$ | 140–260 °C, 3 MPa | n.d. | >99.9% | [88,89] |

Wu et al. tested different supported gold catalyst for MF formation. They could show that first the activation of $CO_2$ and $H_2$ takes place on the gold surface. This is followed by the formation of formic acid and the subsequent esterification with methanol to methyl formate. Thus, the activation of the reactant gases as well as the desorption of MF from the surface are the rate-determining steps of the reaction [87]. These observations are also confirmed by Krocher et al., using a Ru catalyst supported on Si, and by Corral-Pérez et al., using an $Ag/SiO_2$ catalyst [84,88,89].

This approach is an excellent way to capture the emission gas $CO_2$ as a raw material for production of high-value products. However, $CO_2$ is a thermodynamic stable molecule, whereby a high activation energy is necessary. Some studies used $CO_2$ as a supercritical solvent as well as reaction gas. Thus, high quantities of $CO_2$ at high pressure are required. Unfortunately, long reaction times are needed to achieve high yields, which makes implementation in technology difficult from an economic point of view.

### 3.6. Photocatalytic Oxidation of Methanol

The photocatalytic oxidation of methanol has an increasing importance. Titania-based photocatalytic systems are characterized by low cost, easy reuse of the catalytic material and high reactivity. Furthermore, $TiO_2$ is well suitable for photocatalytic oxidation due its remarkable chemical stability with minimum photocorrosion [90].

Czelej et al. could be shown that the bimetallic system $PdAu/TiO_2$ is able to selectively oxidize gaseous methanol to MF. The measured methanol conversion was 80% for $Pd90\text{-}Au10/TiO_2$ as well as $Pd75\text{-}Au25/TiO_2$ and 70% for $Pd50\text{-}Au50/TiO_2$. Experimental investigations as well as DFT calculations demonstrated that the MF selectivity depends strongly on the chemical composition of the metallic support itself. The best photocatalytic performance has been observed for 1:1 $PdAu/TiO_2$ with a selectivity of 70%. A monometallic $Pd/TiO_2$ systems produced only $CO_2$. The authors explained these findings with an

in situ oxidation of palladium to PdO during the photo-oxidation. The Au atoms inhibit this oxidation of Pd and thus keeping the oxidizing power low enough to prevent $CO_2$ formation [91]. According to these results Guo et al. observed only a low MF production via photocatalytic oxidation of methanol with $TiO_2$ without further metal [53].

Li et al. design a new structure of Cu onto $TiO_2$ double layered hollow spheres. In this system, copper species dispersed on the inner surface of the mesoporous titania shell. The highest MF selectivity (77%) could be obtained at stoichiometric ratio (1:0.5) of methanol to oxygen. The methanol conversion increased with increasing oxygen partial pressure up to 85%, while MF selectivity sharply decreased [92].

The photocatalytic oxidation of methanol to MF is an innovative and sustainable approach for MF production. However, photocatalytic oxidation requires targeted irradiation, which is a major cost driver on an industrial scale. Therefore, it is not foreseeable at the present time when large-scale implementation will take place.

### 3.7. Electrolysis of Methanol

A novel approach to the selective production of MF is the electrolysis of methanol. Kishi et al. developed a system with allows the MF production via direct electrolysis of pure methanol at room temperature and atmospheric pressure. The system consists of a membrane electrode assembly (MEA) consisting of Pt/C anode and proton-exchange membranes (PEMs). Within 24 h a TOF of 468 $h^{-1}$ was achieved. Furthermore, it was observable that the MF selectivity increased with decreasing water content because of suppressed $CO_2$ formation. Only from a water content of <20% the formation of $CO_2$ is suppressed [93]. Thus, methanol with a high purity is required for an efficient process, which, together with the electricity costs incurred, makes the process appear very cost-intensive at first glance. Unfortunately, the authors did not make any statement regarding the process costs and the economic efficiency.

### 4. Conclusions and Outlook

Methyl formate is a key component for defossilizing industry and mobility. This C1 building block allows for producing of a wide range of daughter products as well as the utilization as a fuel or fuel additive. For example, the use of methanol as a fuel in internal combustion engines requires the addition of methyl formate to ensure year-round use even at low temperatures. Currently, the industrial production of methyl formate is performed by carbonylation of methanol. However, this process is connected with high requirements on reactant purity and low conversion rates. Hence, new approaches for methyl formate production are sought (Figure 6). Especially, the dehydrogenation and the oxidation of methanol are promising alternatives. However, there is still a great need for research to increase MF yields. A major hurdle is the presence of water in the system. Only an efficient separation of the water will allow an economical production of MF.

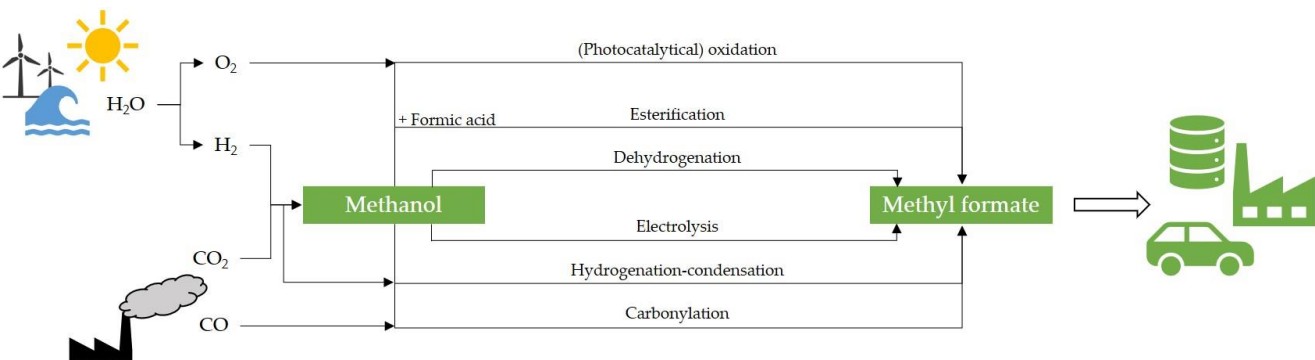

**Figure 6.** Possible synthesis routes of methyl formate from green methanol.

**Author Contributions:** Conceptualization, D.K., L.B. and M.B.; writing—original draft preparation, D.K., L.B. and J.W.; writing—review and editing, M.B., D.K. and L.B.; supervision, D.K.; project administration, D.K. and M.B.; funding acquisition, M.B. All authors have read and agreed to the published version of the manuscript.

**Funding:** The authors thanks SAB (European Social Fund (ESF), grant number 100240621) and VNG for financial support.

**Acknowledgments:** The APC was funded by Technische Universität Bergakademie Freiberg.

**Conflicts of Interest:** The authors declare no conflict of interest.

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
