# Peer review of "Conversion of Green Methanol to Methyl Formate"

_catalysts, doi:10.3390/catal11070869_

Round 1
Reviewer 1 Report
This is a timely and well constructed review, which accounts for the presently used industrial syntheses, as well as the new procedures under recent scrutiny. Indeed, due to the practical importance of methyl formate, a large scale green process for its synthesis is much needed. The review is comprehensive, follows a more-or-less unified structure, well illustrated and contains the appropriate references. Approx. 10% of the references are from the patent literature, which is a welcome feature, concerning the topic of the paper. Unfortunately, the title is a bit misleading, since it gives the impression that there, indeed, are practical greeen ways of manufacturing methyl formate. In fact, what the review describes, are the numerous and ingenious efforts to achieve this goal, nevertheless, it is also seen from the paper, that MF is still produced by applying the century-old technology of methanol carbonylation using fossile-based MeOH and CO. A slight change in the title would be welcome to touch this situation.
There are several small and easy-to-correct mistakes and typos in the paper, and the manuscript needs a thorough check from this aspect. A list of these (propbably not exhaustive) flollows:
l. 48-49 and 56-57: the same sentence is repeated
l. 76: please, define LHV (all other abbrevciations are defined in the text)
l. 104: what is the unit for the number 10 exp(-6)?
l. 134: later, correct: latter
l. 135: heterogenic, correct: heterogeneous
l. 167: electrophile, correct: electrophilicity
l. 193: hypotheses underway(?), perhaps: recently
l. 212: could be observed, correct: could observe
l. 225: The Advantage, The advantage
l. 226: could be shown, correct: could show
l. 227: four times
l. 423: increase due it is for MF decrease, correct: increased (or increases) due to decrease of MF formation (or: on the expense of MF formation)
l. 471: in which, better: with
l. 473: copper
l. 493-494: A computer message in German! Please, correct.
l. 524: so fare, correct: so far; the second part of the sentence repeats the first.
Reviewer 2 Report
My compliments to the authors for a very well structured and written review on methyl formate production. The introduction on the current industrial process, and the alternative processes on methanol dehydrogenation and methanol oxidation demonstrate the solid understanding of the authors in this research area. This review also covers a broad scope, ranging from homogenous catalysts to heterogenous catalysts, and from reaction kinetics and mechanisms to reactor design. This review deserves publication as it is, but perhaps the authors could consider the following suggestions to strengthen the review.
In the abstract (and in the main text), the disadvantages of the industrial methanol carbonylation to methyl formate process are discussed. The use of high pressure and temperature is highlighted to be a disadvantage but this is debatable. Industrially relevant pressure corridor is usually considered to be between 20 to 40 bar, and this is true for syngas/CO2 conversion to methanol (higher pressure required due to thermodynamics) which is the proposed pathway to green methanol. Therefore, the operation conditions of the current industrial process is not a disadvantage from my perspective. A stronger motivation to shift from the current industrial process would be the use of a homogenous catalyst which complicates downstream processing. There is a stronger incentive to shift to heterogenous catalysts, as the authors mentioned as well, and so the methanol dehydrogenation and methanol oxidation processes which utilized heterogenous catalysts are more attractive.
There are a couple of minor formatting/ English errors, e.g. line 225 of page 7, the word ‘Advantage’ should not start with a capital letter. Also, line 492 of page 13 has a sentence in German.
Reviewer 3 Report
The present review article traverses various approaches to the transformation of methanol to methyl formate (MF). Methanol carbonylation, dehydrogenation, and oxidation are specifically focused, and the effect of various catalysts and reaction parameters has been scrutinized. The article is focused on an important topic to produce MF from methanol that will help to achieve a carbon-neutral economy. Although, the authors compiled various aspects of the topic, however, the catalyst synthesis characterization and performance evaluation including mechanism, etc are not discussed. The graphical representation for such processes is missing that will increase the readability of the article. I recommend major revisions. Some comments are:
- The abstract is too condensed and should be elaborated.
- Graphical abstract depicting the relevance of the topic covered in the review must be added.
- Though the authors mentioned the importance of methyl formate in the introduction part, however, how the conversion of methanol to MF can substitute energy/carbon-intensive processes must be explained in the introduction part.
- Graphs, schemes, mechanisms and important findings in some recent articles must be discussed. Additionally, photocatalytic synthesis of MF is gaining momentum, which can be discussed in a separate section. Some recent articles which can be discussed: ACS Appl. Mater. Interfaces2017, 9, 37, 31825–31833; iScience, 2020, 23, 101157; Sci. Technol., 2019, 9, 6240-6252; ACS Omega2019, 4, 1, 1854–1860; ChemCatChem 11.21 (2019): 5269-5274; ACS Sustainable Chemistry & Engineering 8.31 (2020): 11532-11540; ACS omega 5.26 (2020): 15942-15948.
Reviewer 4 Report
The Review proposed by M. Bertau et al. wish to report advances on the synthesis of methyl formate (MF), with a focus on green approaches. Overall I believe the manuscript is well written, and there are not similar report in the literature. However, it lacks in important references which are missing in many parts of the manuscript.
For example, paragraph on homogenous catalysts (Lines 138-204) is very limited in references (only 8) and seems to focus preliminary on reference 4, which is repeated many times.
Line 188 should report references for Cu, Ru, Pt, Pd, Rh, Ir and Au catalysts used for MF formation.
Line 288. Please correct Catalyst with Catalysts
Line 492: an error message in german language has been added in the thext
Line 531. Reference 51 has been added after Reference 52,53 (Line 524). Please correct
Line 509: References on important photocalysts for oxidation of MeOH to MF are missing (e.g. JACS 2013 135 574-577)
In summary, I believe this procedure could be suitbale for publication on Catalysts after few minor revisions and addition of referencees all over the manuscript.
